# Disposal practices of expired and unused medications among households in Mwanza, Tanzania

**Karol Julius Marwa**[1]*, **Glory Mcharo**[2], **Stanley Mwita**[2], **Deogratias Katabalo**[2], **Deodatus Ruganuza**[3], **Anthony Kapesa**[4]

1 Department of Pharmacology, Catholic University of Health and Allied Sciences, Mwanza, Tanzania, 2 School of Pharmacy, Catholic University of Health and Allied Sciences, Mwanza, Tanzania, 3 Department of Parasitology, Catholic University of Health and Allied Sciences, Mwanza, Tanzania, 4 Department of Community Medicine, Catholic University of Health and Allied Sciences, Mwanza, Tanzania

* carol_maro@yahoo.com

**Data Availability Statement:** All relevant data are within the manuscript and its Supporting Information files.

## Abstract

### Background

The community practice towards disposal of expired and unused medications in spite of its adverse impact have been widely neglected in many developing countries. The available guidelines in Tanzania focus on the disposal of expired medications and cosmetics in hospitals and community pharmacies only.

### Aim

The aim of this study was to assess the disposal practice of expired and unused medications at household level in Mwanza city, north-western Tanzania.

### Methodology

The household based cross-sectional study was conducted among 359 randomly selected household members. Semi-structured questionnaires were used for interview during data collection and while STATA® version 13 was used for analysis.

### Results

Out 359 households visited, 252 (70.19%) had medications kept in their houses at the time of data collection. Among them, 10 (4.0%) households had kept medications at their houses because they were still continuing with treatment while 242 (96.0%) kept unused medications which were supposed to be discarded. The main reason for keeping unused or expired medications at home was uncompleted course of treatment (199 (82.20%) after feeling that they had recovered from illness. The main reason for discarding medications were recovering from illness (141(48.7%) and expiry (136 (46.9%). The major discarding practices for medications were disposing into domestic trashes (219 (75.5%) and pit latrines (45 (15.5%). Majority of respondents (273 (76%) were aware that improper disposal of expired medications are detrimental to human health and environment in general.

**Funding:** Authors received no specific funding for
this work.

**Competing interests:** the authors have declared
that no competing interests exist

## Conclusion

Improper disposal of unused and expired medications at household level was a common
practice in the study area. Tailor-made interventions by the Food and Drugs Authority (FDA)
and other national as well as local stake holders are urgently needed to address the
situation.

## Background

Household practice towards disposal of expired or unused medications in most developing
countries including Tanzania lack clear guidelines [1–3]. In Tanzania, the only existing guide-
line established by the Tanzania Medicines and Medical Devices Authority(TMDA) is for dis-
posal of expired or unused medications in drug dispensing units (community pharmacies),
health centers and hospitals [4]. Guidelines for evaluating environmental impact of the exist-
ing and new medications are also lacking in developing countries like Tanzania.

Many medications have been shown to exist in trace amounts in ground water, surface bod-
ies of water and drinking water as a result of improper drug disposal creating a serious concern
[5–7]. Unfortunately, most water treatment plants are designed to filter sediments and bacteria
or viruses but not filtering chemicals or medications [5, 7, 8] which may be hazardous particu-
larly on prolonged use of contaminated water. Most developed countries do monitor the pres-
ence of pharmaceutical wastes or personal care products in waste and /or open water [9]. This
practice is uncommon in developing countries including Tanzania due to non-existence of
locally suitable technology and high economic investment required [8].

Good disposal practices for unused or expired medications at household level include care-
ful containment of the drugs together with other in other inert substances before taking them
to a pharmaceutical waste destruction centre [6, 10, 11]. Incineration though not realistic at
household level is regarded as the best disposal practice for medications [3, 7]. However the
available international guidelines (such as WHO) are intended for national authorities such as
ministries of health, ministries for environment and Drug Authorities neglecting households
level [9, 12]. The implementation of proper disposal practices results into decreased environ-
mental contamination thus decreasing the exposure of the community from drinking water
containing medications [7, 10]. Amid of this, implementation of proper disposal practices
faces many challenges such as lack of standard drug disposal protocols specifically addressing
medications at households level as seen in many countries, some pharmacies refusing to accept
unused and expired medications or discourage the practice and inadequate reinforcement by
FDAs as seen in Tanzania, [3, 13, 14].

Improper drug disposal practice of unused or expired medications is not only associated
with environmental contamination but also risk for accidental poisoning and abuse, wasted
health care resources and lost opportunities for medical treatment, risk for aquatic or wild life
and antimicrobial resistance [15–20]. Improper disposal of medications has been documented
in Iringa Tanzania, and some other countries in Africa including Kenya, Ethiopia and South
Africa as the most used method for medication disposal in household being thrashing into gar-
bage and flushing into toilets [21–24].

Information on medication disposal practices at household level in the study area is lacking
and limited in Tanzania at large. It is therefore important to assess the disposal practices of
medications in households as well as the general public due to the impact it carries. The find-
ings are pivotal for policy making and guide undertaking of appropriate measures. This study

will therefore explore disposal practices of expired and unused medications at household level in Mwanza city, north-western Tanzania.

## Methods

### Study area and subjects

A household based cross-sectional study was conducted from January to August 2015 among 359 respondents from 359 households in Mwanza city, Tanzania. The city has approximately half a million inhabitants, and is located on the southern shores of Lake Victoria [25]. The city was chosen because it is the second largest city in Tanzania and is along Lake Victoria thus exposed to water contamination if there is improper disposal of medications.

### Sampling procedure

Kish Leslie (1965) [26] formula was used to calculate the sample size using an estimated population proportion of 0.13 from a similar study done in Ethiopia [27] to obtain a minimum sample size of 174 respondents. This sample was doubled with the aim increasing the power of the study to a sample size of 348. However, though 368 households were visited nine (9) households were omitted during analysis because their response was not complete thus making a sample size of 359.

Serial sampling method was employed in selecting households from the list given by Local Street leaders where by every 1st, 3rd and 5th households were picked and visited. In each of the selected household, only one member was picked randomly for interview in order to avoid duplication of information as per Kish Leslie [28]. Since the Local Street leaders had a list of all households and their members, therefore the sampled respondents represented the study population in Mwanza.

### Ethical clearance

Ethical clearance (CUHAS 214/2015) was granted by the joint Catholic University of Health and Allied Sciences (CUHAS) and Bugando Medical Centre (BMC) Institutional Review Board. Permission to conduct the study was obtained from the regional administrative secretary in Mwanza. All respondents signed an informed consent. A written consent was also sought from guardians for all respondents who were below eighteen years. The guardians who were grandmothers/fathers felt their grandchildren could give proper information because they were the ones often discarding the medications.

### Data collection

Household members who consented were interviewed using semi-structured questionnaires. The questionnaire was developed in English and translated into the local language (Swahili) as the commonly language used before it was administered to the respondents. Pre-testing of the questionnaire was done at Igogo ward in Mwanza assess practicability and respondent's understanding of the questionnaire. Twenty respondents were interviewed basing on the contents of the questionnaire. The respondent's recommendations regarding the questionnaire were recorded. Minor modifications of the questionnaire were then made. The questionnaire was adapted from a previous study done in Serbia [18].

The questionnaire captured information on socio-demographic characteristics, medications disposal practice, storage of medications and knowledge on impact of inappropriate disposal of medications. Respondents were requested to reveal to the investigator where the drugs were stored before disposal.

## Statistical analysis

Data were analyzed using STATA® 13 (Statistical Corporation, College Station, TX, US). Chi-square tests were performed for determining association between categorical variables where appropriate. P value of ≤ 0.05 was considered as statistically significant.

## Results

Out of 359 households, 252 (70.19%) were having medications at the time of visiting. Only 10 (4.0%) of households were keeping medications because patients were still continuing with the treatment while 242 (96.0%) households had kept unused medications which were supposed to be discarded.

Most households stored medications on cupboards 94 (37.3%) and tins 91(36.1%) as shown on Table 1.

The main reasons for keeping unused or expired medications at home were uncompleted treatment after recovering from their illness reported by 199 (82.20%) respondents, intolerable side effects of the medications 20(8.30%), change in the treatment regime 15 (6.20%) and forgetting to take the medications 8 (3.30%).

The main employed methods for the disposal of medications at household were throwing into domestic trash 219 (59.1%), flashing into the toilet 45 (12.5%) and burning medications 30 (8.4%) as shown in Table 2. Using Pearson's Chi square test, there was no significant difference in distribution of drug disposal practices based on age, sex, occupation and education.

A total of 290 (80.77%) respondents had the experience of discarding medications. The main reason for discarding medications were expiration of medication as reported by 136 (46.9%) respondents, recovering from illness/conditions 141 (48.7%) and notice in color change of the medications 13 (4.4%).

Most of respondents (91.4%) were not aware of the existence of proper medicine disposal methods. Only few as 31(8.6%) had received information from books, media and friends or neighbors (Table 3).

Majority of respondents were aware on the impact of improper disposal of medications particularly to children due to accidental poisoning, environment (plants) and psychiatry patients resulting from poisoning as shown on Table 4.

## Discussion

Majority of households (96%) had kept unused or expired medications which were supposed to be discarded at the time of visiting. This is an alarming threat because possesion of such medications at households may indicate non-compliance and high self self-medication where by patients are most likely not to complete the treatment and administer inadequate doses which is a predisposing factor for development of drug resistance [29].

**Table 1. Storage places for medications at households.**

| Storage | N(%) |
|---|---|
| Tin | 91 (36.1) |
| In a cupboard | 94 (37.3) |
| On a table | 30 (11.9) |
| In a box | 14 (5.5) |
| On top of a fridge | 12 (4.8) |
| In a handbag | 11 (4.4) |
| **Total** | **252 (100)** |

**Table 2. Socio- demographic characteristics of the respondents and their drugs disposal practices at households.**

| Variables | Domestic trashes n (%) | Toilet n (%) | Burn them n (%) | Buried in the ground n (%) | Gave others n (%) | Total n (%) | P-value |
|---|---|---|---|---|---|---|---|
| Male | 54(59.3) | 10(10.9) | 9(9.9) | 1(1.1) | 17(18.7) | 91(100) | **0.46** |
| Female | 165(61.6) | 35(13.1) | 21(7.80) | 0(0) | 47(17.5) | 268(100) | |
| Total | 219(59.1) | 45(12.5) | 30(8.4) | 1(0.3) | 64(17.9) | 359(100) | |
| **Age (in years)** | | | | | | | |
| 10–18 | 10(50.0) | 2(10.0) | 4(20.0) | 0(0.0) | 4(20.0) | 20(100) | **0.12** |
| 19–29 | 128(66.0) | 19(9.8) | 17(8.8) | 1(0.52) | 29(15.0) | 194(100) | |
| 30–65 | 79(56.4) | 24(17.1) | 9(6.4) | 0(0.0) | 28(20.0) | 140(100) | |
| >65 | 2(40.0) | 0(0.0) | 0(0.0) | 0(0.0) | 3(60.0) | 5(100) | |
| Total | 219(59.1) | 45(12.5) | 30(8.4) | 1(0.3) | 64(17.9) | 359(100) | |
| **Education** | | | | | | | |
| Illiterate | 7(41.2) | 2(11.8) | 0(0.0) | 0(0.0) | 8(47.1) | 17(100) | **0.21** |
| Primary | 74(52.9) | 21(15.0) | 13(9.3) | 0(0.0) | 32(22.9) | 140(100) | |
| Secondary | 82(71.9) | 12(10.5) | 7(6.1) | 1(0.9) | 12(10.5) | 114(100) | |
| High School | 9(52.9) | 1(5.9) | 1(5.9) | 0(0.0) | 6(35.3) | 17(100) | |
| College/University | 47(66.2) | 9(12.7) | 9(12.7) | 0(0.0) | 6(8.5) | 71(100) | |
| Total | 219(59.1) | 45(12.5) | 30(8.4) | 1(0.3) | 64(17.9) | 359(100) | |
| **Occupation** | | | | | | | |
| Housemaid | 7(87.5) | 0(0.0) | 0(0.0) | 0(0.0) | 1(12.5) | 8(100) | **0.20** |
| Housewife | 35(53.8) | 8(12.3) | 5(7.7) | 0(0.0) | 17(26.2) | 65(100) | |
| Business person | 93(62.4) | 15(10.1) | 9(6.0) | 0(0.0) | 31(21.5) | 148(100) | |
| Peasant | 14(56.0) | 6(24.0) | 2(8.0) | 0(0.0) | 3(12.0) | 25(100) | |
| Employed | 29(54.7) | 8(15.1) | 9(17.0) | 1(1.9) | 6(11.3) | 53(100) | |
| Student | 33(71.8) | 5(10.8) | 5(8.7) | 0(0.0) | 4(8.7) | 47(100) | |
| Self-employed | 4(57.1) | 3(42.9) | 0(0.0) | 0(0.0) | 0(0.0) | 7(100) | |
| Unemployed | 4(66.7) | 0(0.0) | 0(0.0) | 0(0.0) | 2(33.3) | 6(100) | |
| Total | 219(59.1) | 45(12.5) | 30(8.4) | 1(0.3) | 64(17.9) | 359(100) | |

Furthermore, possession of unused or expired medications at households may lead to exchange of left over medications between household members or neighbours [9] who have similar symptoms to which the drug was prescribed or self-medicated which is common in Tanzania and other developing countries [30].This practice increases the risk for development of drug resistance, treatment failure, abuse, poisoning and toxicity in the community.

The major reason for keeping unused or expired medications at households was uncompleted treatment (82.20%) after subsiding of the symptoms or noncompliance. These findings are in consistence with those recorded from previous studies in USA and other countries [16, 31].

**Table 3. Awareness on existence of proper methods for medications disposal.**

| Source | N (%) |
|---|---|
| Never learned/heard | 328 (91.4) |
| Reading books | 4 (1.1) |
| Seminar(s) | 9 (2.5) |
| Media (Tv& Radio) | 6 (1.7) |
| Neighbor or friend | 12 (3.3) |
| Total | 359(100%) |

**Table 4. Household member's knowledge on the impact of improper disposal.**

| Impact/Effect | N (%) |
|---|---|
| Poisoning to children | 226 63.0) |
| Poisoning to plants and animals | 27 (7.2) |
| Psychiatric people can be poisoned | 22 (6.1) |
| I don't know/no impact | 84 (23.7) |
| **Total** | **359(100)** |

The study has identified cupboards and tins as places where most medications are stored among households (Table 1). Throwing of unsused medications into domestic trashes/garbges which is improper method for disposing medications, was the most common practice for disposal in this study. These findings are similar to those documented in USA, Kuwait, Malaysia, Serbia and many other parts of the world [6, 18, 30–32]. Throwing medications into the toilet particularly latrine pits or flushing down the sink were the second common practices (Table 1) similar to records in USA and Kuwait [31, 33]. Flushing medications down sinks or toilet are regarded as one of the least appropriate methods for disposing medications since it results to transport of medications into water supplies [34].

The improper disposal of unused or expired medications through sinks, dust bins, toilets as recorded in the present study (Table 2) is unfriendly to the environment and may be associated with detrimental effects to the environment [13] and acquatic organisms [19], antimicrobial resistance [35] and detrimental effects to the community health through drinking water [36]. Moreover,this practice may also lead to contamination of vegetables, fruitts and fish which may contain medications in trace amounts. Liquids prepations are the most likely to be added to water systems through rinsing down the sink [13, 37] while solid preparations such as tablets and capsules are mostly likely to be deposited in the rubish bins [13]. Establishment of advanced water treatment technology like reverse osmosis would minimize quantities of dissolved drugs in water supplies in Tanzania.

Studies have shown a correlation between improper disposal of antibiotics and emergence of resistance [38], thus the impact of improper disposal of expired or unused household medications on development of antimicrobial resistance should not be overlooked in Tanzania and other developing countries where self-medication is common and in some of these countries in Africa drugs are sold through the informal economy in open air markets together with vegetables and fruits or by hawkers alongside newspaper vendors [39, 40].

Most of the respondents were unaware on existence of proper methods for disposal of expired medications and had never received any information on proper disposal of medications (Table 3) whereby most of them make uninformed decision in disposing their medications. This is not unexpected since majority of pharmacies do not have consistent recommendation to their customers on medication disposal after completing treatment [33]. The lack of awareness on proper medication disposal methods among household members has also been recorded in developed countries, a few examples being USA [3, 31, 33]. This is due to the fact that little information is passed to the public by FDAs and health care providers on disposal of unused or expired drugs [33, 38, 41]. Surprisingly, majority of respondents were aware on the impact of improper disposal of medications particularly to children due to accidental poisoning, environment (plants) and psychiatry patients resulting from poisoning (Table 4). This is beneficial since studies have a shown most people who know the impact of the practice are likely to participate well in drug disposal programs [13]. The reason why respondents were not aware about existence of proper medication disposal methods but knew about impact of improper medication disposal could be that respondents were referring to

their own method they use in disposing medications and not improper medication disposal in general.

In Tanzania as it is in many other developing countries, hazardous waste centers do not exist and returning of unused or expired medications to pharmacies by the household members or patients is far from reality due to the fact that the TMDA has not put in place the mechanisms or infrastructure for its implementation. Currently there are no guidelines or rules for receiving returned medications from the public and take back programs that accept expired and unused medications from the public/households in Tanzania. A major debate with drug return is the cost associated with it in terms of compensation to patients, pharmacies and other stake holders [34] involved and the fact that even in developed countries such as USA, Sweden and UK only few unused or expired household medications are returned to pharmacies [38, 42]. Therefore this pose a great concern on the practicability of this method in developing countries. Cost-benefit analysis in different countries has confirmed on a need for establishing a pharmaceutical disposal program that involves willingness of manufacturers, pharmacies, distributors, reverse distributors, hospital/clinics, FDAs and even patients to contribute in terms of cost and other requirements [6].

Though the quantity or percentage of active pharmaceutical ingredients from unused or expired medications originating from improper hoseholds disposal is unknown, the existence of improper disposal practices of unused or expired medications as established by the current study gives a brief picture on the possibility of presence of large amounts of unused or expired medications originating from households in our environment.This calls for a well designed and concerted public outreach programme to educate the public on the potential risks associated with the improper medications disposal practices and how to dispose medications properly for the safety of the community.

## Conclusion

Improper disposal of unused and expired medications is a common practice in Mwanza City. There is a little awareness among household members regarding proper ways to dispose unused and expired medicines.

## Supporting information

**S1 File. This is data file of 359 house hold members interviewed in Mwanza,Tanzania.**
(DTA)

**S1 Questionnaire. This is a questionnaire in English and Swahili language which was used to interview house hold members.**
(DOCX)

## Acknowledgments

We sincerely thank all household members who participated in the study and village leaders for their support, the Catholic University of Health and Allied Sciences and regional administrative secretary at Mwanza for granting us a permission to carry the study. We also acknowledge the work of Pendo Ndaki from the department of Development Studies at CUHAS for her editing work.

## Author Contributions

**Conceptualization:** Karol Julius Marwa, Glory Mcharo, Stanley Mwita.

**Data curation:** Glory Mcharo, Deogratias Katabalo, Deodatus Ruganuza.

**Formal analysis:** Karol Julius Marwa, Deodatus Ruganuza, Anthony Kapesa.

**Supervision:** Karol Julius Marwa, Stanley Mwita, Deogratias Katabalo.

**Writing – original draft:** Karol Julius Marwa, Deodatus Ruganuza.

**Writing – review & editing:** Karol Julius Marwa, Glory Mcharo, Stanley Mwita, Deogratias Katabalo, Deodatus Ruganuza, Anthony Kapesa.

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
