## [Decision Letter · Decision Letter 0]

13 Dec 2019

PONE-D-19-25544

Disposal practices of expired and unused medications among households in Mwanza, Tanzania

PLOS ONE

Dear Mr. Marwa,

Thank you for submitting your manuscript to PLOS ONE. After careful consideration, we feel that it has merit but does not fully meet PLOS ONE’s publication criteria as it currently stands. Therefore, we invite you to submit a revised version of the manuscript that addresses the points raised during the review process.

More specifically, pay attention to the followings:

1) Address the grammatical errors possibly by employing professional editing services

2) Strive to draw your arguments from local experiences

3) Address the statistical issues identified by reviewers. 

We would appreciate receiving your revised manuscript by Jan 27 2020 11:59PM. To enhance the reproducibility of your results, we recommend that if <gwmw class="ginger-module-highlighter-mistake-type-3" id="gwmw-15762239745158668547268">applicable you</gwmw> deposit your laboratory protocols in protocols.io, where a protocol can be assigned its own identifier (DOI) such that it can be cited independently in the future. For <gwmw class="ginger-module-highlighter-mistake-type-3" id="gwmw-15762239745193532206479">instructions see</gwmw>: http://journals.plos.org/plosone/s/submission-guidelines#loc-laboratory-protocols

A rebuttal letter that responds to each point raised by the academic editor and reviewer(s). This letter should be uploaded as <gwmw class="ginger-module-highlighter-mistake-type-3" id="gwmw-15762239745487631772864">separate file</gwmw> and labeled 'Response to Reviewers'.A marked-up copy of your manuscript that highlights changes made to the original version. This file should be uploaded as <gwmw class="ginger-module-highlighter-mistake-type-3" id="gwmw-15762239762482169032922">separate file</gwmw> and labeled 'Revised Manuscript with Track Changes'.An unmarked version of your revised paper without <gwmw class="ginger-module-highlighter-mistake-type-3" id="gwmw-15762239769454927620297">tracked</gwmw> changes. This file should be uploaded as <gwmw class="ginger-module-highlighter-mistake-type-3" id="gwmw-15762239776157677572955">separate file</gwmw> and labeled 'Manuscript'.

Please <gwmw class="ginger-module-highlighter-mistake-type-3" id="gwmw-15762239788689576153802">note while</gwmw> forming your response, if your article is accepted, you may have the opportunity to make the peer review history publicly available. The record will include editor decision letters (with reviews) and your responses to reviewer comments. If eligible, we will contact you to opt in or out.

We look forward to receiving your revised manuscript.

Kind regards,

Kahabi Ganka Isangula, MD, MPH, PhD

Academic Editor

PLOS ONE

Additional Editor Comments:

Major revisions required. Kindly pay attention to the followings:

1) Address the grammatical errors possibly by employing professional editing services

2) Strive to draw your arguments from local experiences

3) Address the statistical issues identified by reviewers.

Journal Requirements:

3. Please upload a copy of Supporting Information S1 File and S2 File which you refer to in your text on page 11.

Reviewers' comments:

Reviewer's Responses to Questions

**Comments to the Author**

1. Is the manuscript technically sound, and do the data support the conclusions?

Reviewer #1: Yes

Reviewer #2: Partly

2. Has the statistical analysis been performed appropriately and rigorously? 

Reviewer #1: Yes

Reviewer #2: No

3. Have the authors made all data underlying the findings in their manuscript fully available?

The PLOS Data policy requires authors to make all data underlying the findings described in their manuscript fully available without restriction, with rare exception (please refer to the Data Availability Statement in the manuscript PDF file). The data should be provided as part of the manuscript or its supporting information, or deposited <gwmw class="ginger-module-highlighter-mistake-type-3" id="gwmw-15762239970029267921870">to</gwmw> a public repository. For example, in addition to summary statistics, the data points behind means, medians and variance measures should be available. If there are restrictions on publicly sharing data—e<gwmw class="ginger-module-highlighter-mistake-type-3" id="gwmw-15762239988783898823288">.</gwmw>g. <gwmw class="ginger-module-highlighter-mistake-type-1" id="gwmw-15762239995584732760752">participant</gwmw> privacy or use of data from a third party—those must be specified.

Reviewer #1: Yes

Reviewer #2: Yes

4. Is the manuscript presented in an intelligible fashion and written in standard English?

PLOS ONE does not copyedit accepted manuscripts, so the language in <gwmw class="ginger-module-highlighter-mistake-type-3" id="gwmw-15762240019194595882767">submitted</gwmw> articles must be clear, correct, and unambiguous. Any typographical or grammatical errors should be corrected at revision, so please note any specific errors here.

Reviewer #1: No

Reviewer #2: No

5. Review Comments to the Author

Please use the space provided to explain your answers to the questions above. You may also include additional comments for the author, including concerns about dual publication, research ethics, or publication ethics. <gwmw class="ginger-module-highlighter-mistake-type-3" id="gwmw-15762240086178294661802">(</gwmw>Please upload your review as an attachment if it exceeds 20,000 characters)

Reviewer #1: Reviewer #1: This is a well-conceived study that provides an important assessment of disposal practices of expired and unused medications among households in Mwanza, Tanzania.

INTRODUCTION

1. Page 3 lines 59-60: In Tanzania, the only existing guidelines established by TFDA are for disposal of expired or unused medications at community pharmacies, health centers and hospitals.

The name of the National Medicines Regulatory Authority in Tanzania has been changed from the Tanzania Food and Drug Authority (TFDA) to the Tanzania Medicine and Medical Devices Authority (TMDA). Make the necessary correction.

With respect to lines 59-60, consider referencing the Tanzania guideline.

2. Page 3 lines 60-61: Guidelines for evaluating environmental impact of the existing and new medications are also lacking in developing countries like Tanzania.

Have you checked with the Environment Protection Agency of Tanzania to see whether they have <gwmw class="ginger-module-highlighter-mistake-type-3" id="gwmw-15762240158378498351047">Environmental Impact Assessment</gwmw> (EIA) guideline in this regard?

3. Page 3 lines 68-70: This practice is uncommon practice in developing <gwmw class="ginger-module-highlighter-mistake-type-3" id="gwmw-15762240173961298721530">countries including</gwmw> Tanzania due to <gwmw class="ginger-module-highlighter-mistake-type-1" id="gwmw-15762240173963929588635">nonexistence</gwmw> of suitable technology and high economic investment required [7].

Rewrite sentence and ensure practice is not used twice in the same sentence.

4. Page 3 lines 74-76: Incineration though not realistic at <gwmw class="ginger-module-highlighter-mistake-type-1" id="gwmw-15762240200776598222121">house hold</gwmw> level is regarded as the best disposal practice for medications [3, 6] as advocated by the available international guidelines which in fact themselves are only intended for national authorities neglecting households’ level [8].

Name some of the international guidelines that are only intended for national authorities and examples of some of these authorities? Provide further clarity on this.

5. Page 4 lines 86-88: Improper disposal of medications has been documented in some countries in <gwmw class="ginger-module-highlighter-mistake-type-3" id="gwmw-15762240244361221958633">Africa including</gwmw> Kenya, Ethiopia and South Africa the most used <gwmw class="ginger-module-highlighter-mistake-type-3" id="gwmw-15762240244368848473203">method</gwmw> for medication disposal in household being <gwmw class="ginger-module-highlighter-mistake-type-3" id="gwmw-15762240244363809026537">thrashing</gwmw> into garbage and flushing into toilets<gwmw class="ginger-module-highlighter-mistake-type-3" id="gwmw-15762240244360045096288">[</gwmw>19] [20, 21].

Consider including studies done in Tanzania.

METHODS

1. Page 5 lines 102-103: Kish Leslie (1965<gwmw class="ginger-module-highlighter-mistake-type-3" id="gwmw-15762240270087520024104">)</gwmw>[23] formula was used to calculate the sample size using an estimated population proportion of 0.13 from a similar study done in Ethiopia<gwmw class="ginger-module-highlighter-mistake-type-3" id="gwmw-15762240270088898024160">[</gwmw>24] to obtain a minimum sample of size 174.

Why was sample size not calculated based on similar studies done in Tanzania? <gwmw class="ginger-module-highlighter-mistake-type-3" id="gwmw-15762240281974933153245">Example</gwmw> given (Baltazary, 2013).

2. Page 5 lines 106--111: Serial sampling method was employed in choosing households from the list given by Local Street leaders <gwmw class="ginger-module-highlighter-mistake-type-1" id="gwmw-15762240296841962568113">where by</gwmw> every 1st, 3rd and 5th households were picked and visited. In each of the selected households, only one member was picked randomly for <gwmw class="ginger-module-highlighter-mistake-type-3" id="gwmw-15762240309860017286623">interview</gwmw> in order to avoid duplication of information as per Kish Leslie<gwmw class="ginger-module-highlighter-mistake-type-3" id="gwmw-15762240309868843732302">[</gwmw>25]<gwmw class="ginger-module-highlighter-mistake-type-3" id="gwmw-15762240309862261292766"> .</gwmw> Local Street leaders have a list of all household members in their streets as part of the local government system operating in Tanzania thus were considered to be a reliable source in the sampling process.

Are there not data from a more credible source of statutory statistics organization in Tanzania?

3. Page 5 lines 112-118:

What is the Ethical clearance number?

4. Page 5 lines 119-126:

How was the questionnaire developed? Did you use information from other studies to develop it? If yes, reference those studies where you got the information from. How was the questionnaire pre-tested? Provide more details?

DISCUSSION

1. Ensure that result is discussed chronological as presented in the result

2. You have lots of grammatical errors and spelling mistake in the discussion

Example given:

• Line 177……betwen instead of between

• Line 176…..possesion instead of possession

3. What were the strengths and limitations of your study?

GENERAL COMMENT:

This manuscript could benefit from a professional manuscript editing to improve its grammatical, spelling and intellectual clarity.

This manuscript could benefit from a professional manuscript editing to improve its grammatical, spelling and intellectual clarity.

Reviewer #2: The article must be revised and remove all the grammatical errors and address the following points. 1. Apply more statistical tools form a person having command on statistics

2. adopt uniform format for tables.

6. PLOS authors have the option to publish the peer review history of their article (what does this mean?). If published, this will include your full peer review and any attached files.

Reviewer #1: No

Reviewer #2: No

<gdiv></gdiv>

---

## [Author Response · Author response to Decision Letter 0]

10 Jul 2020

Title: Disposal practices of expired and unused medications among households in Mwanza, Tanzania.

Department of pharmacology, Catholic university of Health and Allied Sciences, Mwanza, Tanzania

Email:carol_maro@yahoo.com

13th June 2020

Editor, PLOS ONE

Dear editors, 

 Please find the edited manuscript. The reviewer’s comments have been addressed as shown below.

No. Reviewer’s/editor’s comment Response

1. Please upload a copy of Supporting Information S1 File and S2 File which you refer to in your text on page 11. The S1 and S2 files have been uploaded as suggested.

2. The conclusions must be drawn appropriately based on the data presented.

 The conclusion has been re-written basing on data presented 

3. The name of the National Medicines Regulatory Authority in Tanzania has been changed from the Tanzania Food and Drug Authority (TFDA) to the Tanzania Medicine and Medical Devices Authority (TMDA). Make the necessary correction. Editing has been done by replacing Tanzania Food and Drug Authority (TFDA) with TMDA as suggested by the reviewer 1.

4. With respect to lines 59-60, consider referencing the Tanzania guideline. The Tanzania guideline has been referenced as required

5. Page 3 lines 68-70: Rewrite sentence and ensure practice is not used twice in the same sentence. Rewriting of the sentence has been done as suggested. The word practice has is not repeated.

6. Page 3 lines 74-76: Name some of the international guidelines that are only intended for national authorities and examples of some of these authorities? Provide further clarity on this Clarification has been done by mentioning WHO and listing the authorities as shown in the corrected manuscript. The WHO guideline has been inserted as a reference.

7. Page 4 lines 86-88: Improper disposal of medications has been documented in some countries in Africa including Kenya, Ethiopia and South Africa the most used method for medication disposal in household being thrashing into garbage and flushing into toilets [ 19] [20, 21]. Consider including studies done in Tanzania. A similar study done in Iringa Tanzania has been referenced.

8. Page 5 lines 102-103: Kish Leslie (1965 ) [23] formula was used to calculate the sample size using an estimated population proportion of 0.13 from a similar study done in Ethiopia [ 24] to obtain a minimum sample of size 174.Why was sample size not calculated based on similar studies done in Tanzania? Example given (Baltazary, 2013). Authors admit to have overlooked Baltazary et al 2013 by the time we were calculating the sample size. However, using the study as a reference to calculate the sample size we get 188 participants which is about a half of the participants enrolled in our study (359 participants). It is obvious not using baltazary etal has no effect as far as the quality of our data is concerned. 

9. Local Street leaders have a list of all household members in their streets as part of the local government system operating in Tanzania thus were considered to be a reliable source in the sampling process. Are there not data from a more credible source of statutory statistics organization in Tanzania? The only available data at district level is the total number of people in each street. This is the reason why local street leaders who have a list of household members in their streets as part of the local government system are considered to be a reliable source of information. Even the central government in Tanzania obtains its primary data from the local authorities. 

10. Page 5 lines 112-118:

What is the Ethical clearance number? The ethical clearance has now been mentioned as seen in the manuscript 

11. Page 5 lines 119-126: How was the questionnaire developed? Did you use information from other studies to develop it? If yes, reference those studies where you got the information from. How was the questionnaire pre-tested? Provide more details? More details on the pre-testing and developing of the questionnaire s now provided as seen in the manuscript 

12. Ensure that result is discussed chronological as presented in the result The discussion is now in chronological order. Tables have been re-arranged.

13. You have lots of grammatical errors and spelling mistake in the discussion Example given:

• Line 177……betwen instead of between

• Line 176…..possesion instead of possession Grammatical errors and spelling mistakes have been addressed as suggested by the reviewer. Changes are seen in the new manuscript with track changes. 

14. Reviewer 2. The article must be revised and remove all the grammatical errors. Grammatical errors have been addressed as stated above 

15. Apply more statistical tools form a person having command on statistics Authors went through the primary data, STATA commands and tables were cross checked 

16. Adopt uniform format for tables Reviewers suggestions have been taken into account

17. 

 I would like to thank the reviewers and the editors for their valuable comments which have made this manuscript better than before. 

Please contact me if any further clarification is needed. 

Sincerely yours,

Karol J. Marwa

Corresponding author.

---

## [Editor Report · Decision Letter 1]

10 Aug 2020

PONE-D-19-25544R1

Disposal practices of expired and unused medications among households in Mwanza, Tanzania

PLOS ONE

Dear Dr. Marwa,

Thank you for submitting your manuscript to PLOS ONE. After careful consideration, we feel that it has merit but does not fully meet PLOS ONE’s publication criteria as it currently stands. Therefore, we invite you to submit a revised version of the manuscript that addresses the points raised during the review process.

The specific actions needed before accepatance are:

A number of grammatical errors can be seen in the document. Consider engaging professional proofreading & editing services and provide appropriate acknowledgement.Fully attend to reviewer's comments on ensuring that the discussion section is chronologically presented in line with presentation of the results. 

We look forward to receiving your revised manuscript.

Kind regards,

Kahabi Ganka Isangula, MD, MPH, PhD

Academic Editor

PLOS ONE

Additional Editor Comments (if provided):

1. A number of grammatical errors can be seen in the document. The authors need to engage professional editing services and provide acknowledgement.

2. Reviewers comments on discussing the results in chronological order in line with result presentation is weakly attended to.

<gdiv></gdiv>

---

## [Author Response · Author response to Decision Letter 1]

28 Dec 2020

No. Reviewer’s/editor’s comment Response

1. A number of grammatical errors can be seen in the document. Consider engaging professional proofreading & editing services Grammatical errors have been corrected as suggested by the editor

2. Fully attend to reviewer's comments on ensuring that the discussion section is chronologically presented in line with presentation of the results The discussion is now in line with the results presentation. The order has been addressed. 

3.

---

## [Decision Letter · Decision Letter 2]

20 Jan 2021

Disposal practices of expired and unused medications among households in Mwanza, Tanzania

PONE-D-19-25544R2

Dear Dr. Marwa,

We’re pleased to inform you that your manuscript has been judged scientifically suitable for publication and will be formally accepted for publication once it meets all outstanding technical requirements.

Kind regards,

Kahabi Ganka Isangula, MD, MPH, PhD

Academic Editor

PLOS ONE

Additional Editor Comments (optional):

Reviewers' comments:

Reviewer's Responses to Questions

**Comments to the Author**

1. If the authors have adequately addressed your comments raised in a previous round of review and you feel that this manuscript is now acceptable for publication, you may indicate that here to bypass the “Comments to the Author” section, enter your conflict of interest statement in the “Confidential to Editor” section, and submit your "Accept" recommendation.

Reviewer #2: All comments have been addressed

Reviewer #3: All comments have been addressed

2. Is the manuscript technically sound, and do the data support the conclusions?

Reviewer #2: Yes

Reviewer #3: Yes

3. Has the statistical analysis been performed appropriately and rigorously? 

Reviewer #2: Yes

Reviewer #3: Yes

4. Have the authors made all data underlying the findings in their manuscript fully available?

Reviewer #2: Yes

Reviewer #3: Yes

5. Is the manuscript presented in an intelligible fashion and written in standard English?

Reviewer #2: Yes

Reviewer #3: Yes

6. Review Comments to the Author

Reviewer #2: The reviewers have answered all the questions raised by the reviewers. The article is accepted for pubilcation in Plos One.

Reviewer #3: The reviewers have answered all the questions of the reviewers and the article is accepted for possible publication in the journal.

7. PLOS authors have the option to publish the peer review history of their article (what does this mean?). If published, this will include your full peer review and any attached files.

Reviewer #2: No

Reviewer #3: No

<gdiv></gdiv>

---

## [Editor Report · Acceptance letter]

25 Jan 2021

PONE-D-19-25544R2 

Disposal practices of expired and unused medications among households in Mwanza, Tanzania. 

Dear Dr. Marwa:

I'm pleased to inform you that your manuscript has been deemed suitable for publication in PLOS ONE. Congratulations! Your manuscript is now with our production department. 

Kind regards, 

on behalf of

Dr. Kahabi Ganka Isangula 

Academic Editor

PLOS ONE